# Proline-Specific Fungal Peptidases: Genomic Analysis and Identification of Secreted DPP4 in Alkaliphilic and Alkalitolerant Fungi

**DOI:** 10.3390/jof7090744

**Published:** 2021-09-10

**Authors:** Nikita Alkin, Yakov Dunaevsky, Elena Elpidina, Galina Beljakova, Valeria Tereshchenkova, Irina Filippova, Mikhail Belozersky

**Affiliations:** 1Biological Faculty, Lomonosov Moscow State University, 119991 Moscow, Russia; nikita9801@mail.ru (N.A.); adm-odo@yandex.ru (G.B.); 2A.N. Belozersky Institute of Physico-Chemical Biology, Lomonosov Moscow State University, 119991 Moscow, Russia; elp@belozersky.msu.ru (E.E.); mbeloz@belozersky.msu.ru (M.B.); 3Chemical Faculty, Lomonosov Moscow State University, 119991 Moscow, Russia; lerunechka_lu@mail.ru (V.T.); irfilippoff@yandex.ru (I.F.)

**Keywords:** ascomycetes, alkaliphilic and alkali-tolerant fungi, basidiomycetes, bioinformatic analysis, proline-specific peptidases, secretion

## Abstract

Proline-specific peptidases (PSP) play a crucial role in the processing of fungal toxins, pheromones, and intracellular signaling. They are of particular interest to biotechnology, as they are able to hydrolyze proline-rich oligopeptides that give a bitter taste to food and can also cause an autoimmune celiac disease. We performed in silico analysis of PSP homologs in the genomes of 42 species of higher fungi which showed the presence of PSP homologs characteristic of various kingdoms of living organisms and belonging to different families of peptidases, including homologs of dipeptidyl peptidase 4 (DPP4) and prolyl aminopeptidase 1 found in almost all the studied fungal species. Homologs of proliniminopeptidases from the S33 family absent in humans were also found. Several studied homologs are characteristic of certain taxonomic groups of fungi. Phylogenetic analysis suggests a duplication of ancestral DPP4 into transmembrane and secreted versions, which predate the split of ascomycete and basidiomycete lineages. Comparative biochemical analysis of DPP4 in alkaliphilic and alkali-tolerant strains of fungi showed that, notwithstanding some individual features of these enzymes, in both cases, the studied DPP4 are active and stable under alkaline conditions and at high salt concentrations, which makes them viable candidates for biotechnology and bioengineering.

## 1. Introduction

Fungi occupy various ecological niches, owing to their remarkable physiological plasticity. For example, several fungal species thrive in extreme temperatures and harsh radiation [1,2,3,4]. The extremophilic groups of fungi also include fungi of alkaline habitats that can grow and develop at a pH of 10 or higher [5,6]. The survival of fungi depends on their ability to use nutrients from living or dead organic materials. The encoded set of enzymes largely determine the habitat these fungi can occupy. Thus, differences in secreted enzymes can significantly affect their ability to colonize plants and animals. The peptidases secreted by fungi, performing trophic functions, cleave the bonds in protein substrates and lead to the formation of amino acids and peptides, which are absorbed by fungal cells and used in catabolic and biosynthetic processes.

Secreted peptidases from alkaliphilic fungi are of significant interest to biotechnology [7]. The turn of the 20th and 21st centuries saw expanding practical use of alkaliphilic bacterial enzymes, which served as a starting point for research on the physiology and biochemistry of fungi in alkaline habitats. Among the currently known groups of alkaliphilic and alkali-tolerant fungi, the family Plectosphaerellaceae takes a special place [8]. Representatives of this family have a wide range and are described in detail both taxonomically and physiologically; the genome of one of them (*Sodiomyces alkalinus*) is fully sequenced [9]. However, little is known about the biochemistry of these fungi. The diversity and functioning of the enzymes in these fungi, in particular, to date, highly specific enzymes, such as proline-specific peptidases (PSP), have not been analyzed in this family of fungi.

The proline imino group, unique to proteinogenic amino acids, is a part of the pyrrolidine ring, which renders the peptide bonds formed by proline amino acid residues highly resistant to the effects of classical peptidases. Proteins with a high proline content and proline-containing peptides are among the most significant for metabolism, nutrition, and cellular recognition, in particular, of intracellular signal transmission [10]. Many structural and storage proteins and peptides contain proline, the physiological need for which remains high throughout the life cycle [11]. PSPs are produced by bacteria, archaea, and eukaryotes to break down proline-rich proteins and peptides that are difficult to hydrolyze. Apart from them, two nonspecific enzymes, leucine aminopeptidase (LAP) and cytosolic nonspecific dipeptidase (CND), are able to cleave off the N-terminal Pro.

The practical interest in PSP is largely associated with their possible application to produce gluten-free products, as well as medicines to treat certain gastrointestinal diseases. Among the storage proteins of monocotyledonous seeds used for food by humans and animals, a large group of proteins has a high proline content (more than 30%). These are the so-called gluten proteins (in particular, prolamins), which are often the main food source, but the proline-rich peptides formed from them during incomplete digestion cause the autoimmune celiac disease in sensitive people. The use of commercially available PSPs from various sources to neutralize the immunogenic potential of proline-rich gluten peptides could be one of the main applications of their unique specificity and an attractive option for patients with celiac disease. Oral enzyme therapy using such gluten-destroying enzymes appears to be a promising therapeutic approach. While the PSPs of bacteria [12] and animals [13] are characterized in detail, the data on the PSP of fungi are limited to studies of prolyl endopeptidase from *Aspergillus niger* [14], *Flammulina velutipes* [15], and *A. oryzae* [16].

The aim of this work is to evaluate the potential of fungi in the production of various types of PSP using a bioinformatic approach, followed by the biochemical identification of PSP in the culture of fungi, as well as to characterize their properties that determine the possibility of their further use. Fungi growing under extreme conditions are of scientific interest, both for studying the adaptive evolution of fungi and for evaluating their potential as producers of commercially valuable enzymes. Therefore, we tried to use representatives of the Plectosphaerellaceae family, on the one hand, as a source of enzymes capable of cleaving complex proline-containing bonds, and on the other hand, we tried to compare the production of enzymes and their properties in two groups of fungi—alkaliphiles and alkali-tolerants.

## 2. Materials and Methods

### 2.1. Materials

Culture medium components—malt extract (Maltax 10), (Senson, Niemenkatu, Finlandia), yeast extract (Biomedicals LLC, Santa Ana, CA, USA), casein (Merck, Darmstadt, Germany), peptone (TM-Media, Delhi, India); substrate Ala-Pro-*p*Na was synthesized according to standard procedures [17] in the Protein Chemistry Laboratory, Faculty of Chemistry, Moscow State University; inhibitors—PMSF(Serva, Heidelberg, Germany), diprotin A and diprotin B (Bachem, Bubendorf, Switzerland), sitagliptin and vildagliptin (BioVision, Milpitas, CA, USA), ethylenediaminetetraacetic acid (EDTA) (Merck, Darmstadt, Germany), iodoacetamide (Merck, Darmstadt, Germany).

### 2.2. In Silico PSP Analysis

*Analyzed peptidases*:

Prolyl oligopeptidase (POP), prolidase (XPD), fibroblast activation protein (FAP), dipeptidyl peptidases (DPP)4, DPP6, DPP8, DPP9, DPP10, X-Pro aminopeptidases (APP)1 and APP2, acid prolyl endopeptidase (EPR) from *Aspergillus niger*, prolyliminopeptidases from *Aeromonas sobria* (PIP1), from *Lactobacillus delbrueckii* (PIP2), and from carrot (*Daucus carota*, PIP3), leucine aminopeptidase (LAP), and cytosolic nonspecific dipeptidase (CND).

A total of 42 species of dikaryomycetes with a sequenced genome, including the *S. alkalinus* alkaliphile (Table 1), were selected from the NCBI database. The presented set of species includes representatives of all 6 subdivisions of higher fungi, as well as representatives of various morphological groups (yeast, mycelial fungi, lichenized fungi) and ecological-trophic groups (humus saprotrophs, xylotrophs, phytopathogens, entomopathogens). In each of the selected genomes, homologs of the known 12 human PSP (CND—NP_001161971.1 (presents the sequence numbers from GenBank that were used as a request), LAP—NP_056991.2, POP—NP_002717.3, XPD—AAA60064.1, FAP—NP_001278736.1, DPP4—NP_001366533.1, DPP6—NP_001034439.1, DPP8—XP_016877870.1, DPP9—NP_001371540.1, DPP10—NP_001171507.2, APP1—XP_016872104.1, and APP2—NP_003390.4), as well as those of acid EPR—XP_025457004.1 from *Aspergillus niger*, of PIP1 from *Aeromonas sobria*—BAA06380.1, from *Lactobacillus delbrueckii* (PIP2)—AAA61596.1, and from carrot (*Daucus carota*, PIP3—KZM82276.1), were searched by amino acid sequences using the Protein BLAST service [18]. The cut-off threshold was E-value 1 * 10–20 and 75% coverage. To improve the homology attribution accuracy of the detected sequences, clustering of all sequences using the EFI-EST service was used [19]. The following parameters were further calculated for each homolog: molecular weight, number of amino acids, and isoelectric point using the online resource Isoelectric Point Calculator [20]; presence, size, and position of the signal peptide; and transmembrane domain (Signal P-5.0 and TMHMM Server V. 2.0 services) [21]. After that, the amino acid sequences of the detected PSP homologs were aligned using the MEGA X package. In the Batch CD-Search service [22], a search was performed for conservative domains of the PSP homologs. For the most common DPP4 peptidase, a phylogenetic tree of homologs was constructed using the MEGA X ML package. The DPP4 amino acid sequence of humans (*Homo sapiens*) was selected as an outgroup.

### 2.3. Fungal Strains

Five species of alkaliphilic and alkali-tolerant ascomycetes from the family Plectosphaerellaceae (Glomerellales; Sordariomycetes) and one alkali-tolerant species from the family Hypocreaceae (Hypocreales; Sordariomycetes) were studied. Each species was represented by one strain (*Sodiomyces magadiensis* was represented by two strains), obtained from the collection of the Department of Mycology and Algology of the Lomonosov Moscow State University. The cultures are listed in Table 2.

### 2.4. Culture Media

We used alkaline malt agar of standard composition, as well as three modifications of alkaline malt agar to produce liquid media different in the protein source [8]. The mineral component of these media in all three modifications was identical to that of alkaline malt agar. The composition of the nutrient component, in addition to the yeast extract, included either malt extract (15° on the Balling scale), casein, or peptone.

Two components were sterilized and mixed similarly to the preparation of the alkaline malt agar. The mineral and nutrient components were sterilized for 1 h in an autoclave at 0.5 ATI, and mixing was performed in a flask with the nutrient component in a sterile laminar box at 50–60 °C.

### 2.5. Obtaining Submerged Cultures

The fungal cultures were stored on slanted alkaline malt agar in a cold room at 4 °C. In a laminar box, agarized blocks with surface mycelium were transferred from slanted agar (alkaline malt agar) to 150 mL Erlenmeyer flasks containing 50 mL of liquid alkaline fermentation medium and placed on an orbital shaker at 25 °C; the rotation speed was 250 rpm.

### 2.6. Measuring the Enzymatic Activity of Proline-Specific Peptidase

The activity of PSP was measured spectrophotometrically. A total of 175 µL of the universal buffer (UB) with a given pH [23] was placed in a well of a 96-well polystyrene plate for enzyme immunoassay, then 20 µL of the test filtrate and 5 µL of a chromogenic substrate with a p-nitroanilide label (λ = 405 nm) were added. The total volume of the reaction mixture in each well of the plate was 200 µL. In the control wells, the culture fluid was replaced with 20 µL of UB. Plates with the reaction mixture were incubated in an ELx800 spectrophotometer at 37°C for 45 to 120 min in the mode of kinetic analysis of the optical density (λ = 405 nm); the optical density of the reaction mixture was measured every 5 or 10 min, starting from the moment of adding the substrate (zero moment, which was considered as negative control). Gen5 BioTek software was used to work with the spectrophotometer and the obtained data were processed in MO Excel 2018 program.

We used a chromogenic substrate with a *p*-nitroanilide label at the C-end that is specific to dipeptidyl peptidase 4 (DPP4), namely, Ala-Pro-*p*Na. The substrate was dissolved in dimethylformamide at a concentration of 10 mM. The amount of the enzyme per 1 mg of dry mycelium mass, which, when the substrate was hydrolyzed under the specified incubation conditions, increased the optical density of the solution by 0.01 in 1 h at 405 nm, was taken as a unit of enzymatic activity. To obtain dry biomass, the filter paper with mycelium was placed in an oven at a temperature of 80 °C for 2 h, after which it was cooled and weighed; the procedure was repeated until a constant weight was reached. The dry mass of the mycelium was calculated by subtracting the mass of the control paper filter that was pre-weighed and subjected to a similar incubation in an oven.

### 2.7. Optimizing the Culture Medium and Mycelium Cultivation Period

Mycelia of *S. alkalinus* F11, *S. magadiensis* B39, and *S. tronii* MAG3 were grown for 10 days in three alkaline fermentation media containing three alternative protein sources (barley malt, casein, and peptone). When the mycelium was formed, the culture fluid filtrates were prepared and the activity of DPP4 was measured. The peptone medium with the highest DPP4 activity was used in further experiments. To determine the optimal cultivation period, these strains were grown on the peptone culture medium for 4, 7, 11, 14, 18, 19, 21, and 24 days, after which the culture fluid filtrates were prepared and DPP4 activity was measured. The culture filtrate was separated from the mycelium after incubation on an orbital rocker by double filtration through filter paper, after which the filtered liquid was repurified by centrifugation (6000 rpm, 10 min). To prevent biological contamination, 8% sodium azide (2.5 µL/mL NaN_3_) was added to the resulting filtrate. The filtrates were stored at 4 °C.

### 2.8. Determination of the pH Activity Optimum of DPP4

A total of 20 µL of the solution containing the test PSP (14-day culture fluid filtrate), 175 µL of UB of the selected acidity to obtain the required pH (3.1–9.5), and 5 µL of the chromogenic substrate were added to the well of a polystyrene plate. The resulting reaction mixture was analyzed spectrophotometrically following the procedure described above.

### 2.9. Determiniation of the pH Stability of DPP4

A total of 20 µL of the solution containing the test PSP (14-day culture liquid filtrate) and the given volume of UB with the selected acidity were added to the well of a polystyrene plate to obtain the required pH (2.0–13.0). The resulting solution was incubated for 60 min with constant stirring on a shaker (300 rpm); then, UB with the selected pH was added to the volume of 195 µL to obtain a pH of the incubation mixture of 7.5 (the optimal pH for the PSP from the previous experiment). Then the resulting mixture was mixed, 5 µL of chromogenic substrate was added, and kinetic analysis of the enzymatic reaction was performed using a spectrophotometer.

### 2.10. Determination of PSP Stability in the Presence of NaCl

A total of 175 µL of UB (pH 7.5) with varying NaCl concentrations: 2 M, 3 M, 4 M, 5 M, and 6 M were added to 20 µL of the test solution. After adding the buffer, the enzymatic reaction was started using 5 µL of the chromogenic substrate, and the kinetics of the colored product formation was observed using a spectrophotometer. The same reaction mixture without NaCl served as a control.

### 2.11. Studying the Effect of Peptidase Inhibitors on PSP Activity

A total of 20 µL of the culture filtrate, 170 µL of the universal buffer (pH 7.0), and 5 µL of the inhibitor stock solution were initially placed in the well of the plate. The resulting mixture was incubated at room temperature for 20 min, after which 5 µL of the chromogenic substrate was added and the kinetic analysis of the enzymatic reaction was performed.

All experiments were carried out in 3 replicates. Average values of peptidase activities and standard deviations were calculated.

## 3. Results and Discussion

### 3.1. Bioinformatic Analysis of Amino Acid Sequences of PSP of Higher Fungi

To determine the prevalence of PSP homologs in dikaryomycetes, 42 genomes of various ascomycetes and basidiomycetes were analyzed. In total, 10 different PSPs were found in the species, including homologs of human APP1, CND, DPP4, LAP, prolyl oligopeptidase (POP), and prolidase (XPD). Homologs of DPP4 and APP1 were found in the genomes of almost all selected species, suggesting their important functional significance, while homologs of fibroblast activation protein (FAP), DPP6, DPP8, DPP9, DPP10, and APP2 were not found in the fungal genomes (Table 1).

Some of the detected homologs were shown to specifically belong to certain taxa, which allows us to consider them as potential markers. Thus, LAP and POP homologs were found in the genomes of all the studied basidiomycete species, while, among ascomycetes, only 2 of the 32 analyzed species had such homologs, namely, the entomopathogenic mycelial fungus *Beauveria bassiana* and the soil yeast *Saitoella complicata*. Notably, taxonomically similar representatives, such as *Cordyceps confragosa* and *Metarhizium anisopliae* in the first case, and *Taphrina deformans* in the second one, lacked homologs of both these enzymes. Another typical example of the taxonomic specificity of PSP was the absence of EPR homologs of *A. niger* in the genomes of representatives of the Saccharomycotina subdivision that have a yeast life form or mycelial–yeast dimorphism. All the studied fungal species had homologs of human DPP4.

Out of the three different forms of prolyliminopeptidase (PIP) found in the genomes of higher fungi, PIP1, homologous to the *A. sobria* enzyme, was found in 28 species of ascomycetes and agaricomycetes, while rust and smut fungi lacked this type of PIP. PIP1 is a classical serine peptidase consisting of a single domain of the alpha/beta hydrolase superfamily. PIP2, homologous to the *L. delbrueckii* enzyme, occurred in six ascomycete species in one copy, and in four agaricomycete genomes in several copies (up to eight in the *Daedalea quercina* genome). PIP2 also consists of a single domain, Pro_imino_pep_2, with no signal peptide or transmembrane regions. PIP3 homolog of *D. carota* was found in all studied basidiomycetes (including rust and smut fungi) and only in four representatives of ascomycetes. The domain structure of PIP3 coincides with that of PIP1; the signal peptide and transmembrane regions were also absent.

In some cases, the data indicate the environmental specificity of the PSP. Thus, prolidase (XPD) homologs were found in 41 of the 42 genomes analyzed, while the absence of such homologs is characteristic of entomopathogenic fungus *Hypocrella siamensis* that specifically infects scale insects and whiteflies. Other examples of PSP ecological specificity include the multicopy nature of PIP homologs in wood-degrading basidiomycetes (for example, in *Daedalea quercina* and *Calocera cornea*).

Active sites of detected PSP are largely conserved between humans and fungi. The homologs of serine PSPs retain a conservative triad of amino acid residues, and the homologs of metallopeptidases have binding sites with metal ions and substrates. In contrast to insects, the studied fungi lack inactive homologs, i.e., homologs with substitution in the catalytic triad. Figure 1 shows an example of the alignment of conservative DPP4 sites in representatives of various subdivisions of higher fungi in relation to human DPP4.

The isoelectric points of most of the identified PSP homologs calculated using the online resource Isoelectric Point Calculator are in the range of 5–6. The exceptions are EPR homologs, one third of which, according to the calculations, are characterized by charge loss at pH 4–5, and APP1 homologs, half of which have an isoelectric point value of 8–9. For example, similar APP1 homologs are typical of all the studied species of the Agaricomycotina subdivision. Since the majority of the found PSP homologs possess neither a signal peptide, nor a transmembrane site, nor a nuclear localization signal, they presumably have either a cytoplasmic localization or at least a cytoplasmic transport pathway. Notably, if two DPP4 homologs are present in ascomycetes, one of them often contains a signal peptide sequence, and the second one contains a transmembrane domain. These homologs probably have different localization and perform different functions, i.e., they are paralogs.

To assess the evolutionary relationship of DPP4, we reconstructed a phylogenetic tree of its homologs (Figure 2). In this tree two distinct groups of homologs differing in the presence of a signal peptide can be distinguished. Secreted forms are found exclusively in the Pezizomycotina subdivision in the Ascomycota division, while transmembrane forms are found in all six subdivisions of higher fungi. Transmembrane homologs, in turn, are grouped according to the taxonomic position of objects into clades corresponding to the divisions Basidiomycota and Ascomycota, and further into subdivisions within these clades. Thus, the duplication of the DPP4 gene may have occurred in the early evolution of the Pezizomycotina group, resulting in subsequent diversification of paralogs into transmembrane and secreted forms. Furthermore, this phylogenetic tree explains why FAP homologs are absent in the analyzed fungal genomes. Based on the available data, the DPP4 and FAP enzyme evolutionary lines might have parted after the separation of the fungal and animal evolutionary lines, which is manifested in the close clustering of the human and bovine FAP orthologs, as well as the human and porcine DPP4 orthologs. Thus, fungal homologs of DPP4 can be considered as homologs of the common ancestral enzyme for FAP and DPP4.

The bioinformatic analysis of the whole genome of the alkaliphile *S. alkalinus* F11 conducted using the Protein BLAST service showed the presence of nine homologs of the known PSPs found in representatives of various kingdoms of nature (humans, plants, fungi, and bacteria), including two homologs capable of cleaving proline CND (Table 1). Thus, according to the data obtained, *S. alkalinus* contained two homologs of the DPP4 gene, the larger one of which (ROT37187.1) contained a fragment encoding the transmembrane region, while the shorter one (ROT38925.1) lacked this motif but encoded an N-terminal signal peptide. The calculated isoelectric points of the DPP4 protein homologs of this strain lay close to each other (4.80 and 4.66), but the length of the molecules differs by 145 amino acids, and their masses are 106.0 and 87.5 kDa, respectively. Both homologs contained the conservative N-terminal domain DPP_IV in the first third of the amino acid sequence and the prolyl oligopeptidase domain Peptidase_S9 near the C-terminus. The comparison of the active centers of these homologs with the active center of human DPP4 did not reveal any strong functional differences in the amino acid environment of the conservative triad of serine, aspartic acid, and histidine residues characteristic of serine peptidases.

In *S.alkalinus*, the potential homolog of acidic prolyl endoprotease of *A. niger* (ROT41790.1) had an estimated molecular weight of 44.8 kDa and an isoelectric point characteristic of homologs of this enzyme being equal to 4.6. No specific conservative domains were found in this homolog, but the active center contained a conservative triad of amino acid residues. The amino acid sequence of this homolog was devoid of a signal peptide and transmembrane sites, as well as of a nuclear localization signal, which suggested its localization in the cytoplasm.

One homolog of human aminopeptidase P1 (ROT40160.1) was also found in the genome of *S. alkalinus* F11. As this homolog lacked transmembrane sites and a signal peptide, it is most likely to be localized in the cytoplasm. The APP1 homolog and its ortholog in the human body have all the amino acid residues involved in the co-ordination of manganese ions in the active center of enzymes, which suggested the enzymatic function of this homolog. The isoelectric point of the homolog was 5.7, and the estimated molecular weight was 68.6 kDa. The conserved domains found in this sequence are typical of peptidases from the M24 family (the AMP_N structural domain at the N-end of the molecule and the domain with the prolidase reaction center at the C-end).

The same family of metallopeptidases included three homologs of human prolidase (ROT42762. 1, ROT34876. 1, and ROT38161. 1), whose molecular weights were 56.9, 49.6, and 49.6 kDa, respectively. As in the previous case, SignalP and TMHMM analyses did not detect signal peptides and transmembrane regions in these sequences, and active centers of these homologs were not affected by mutations. The isoelectric points of XPD homologs were also in the slightly acidic range (5.6–5.7). The detected prolidase homologs showed the presence of conservative AMP_N and prolidase domains.

Two of the peptidase homologs found in silico and able to cleave proline bonds (ROT43826. 1 and ROT41757.1) were homologs of human cytosolic nonspecific dipeptidase. Of note, despite the twofold difference in molecular mass (100.6 kDa and 52.2 kDa, respectively), all the conserved amino acid residues important for catalysis were found in both homologs. The heavier CND homolog appeared to contain a large N-terminal insertion about 470 amino acid residues long, with no specific homology with known conservative domains. Despite the significant differences in the peptide chain length, both homologs had similar isoelectric points (5.32 and 5.21, respectively). No signal peptide or transmembrane sites were found in these homologs.

### 3.2. Biochemical Identification of PSP from Alkaliphilic and Alkali-tolerant Fungi

Identification and study of the secreted PSPs were performed on five species of alkaliphilic and alkali-tolerant mycelial dikaryomycetes from the family Plectosphaerellaceae (Glomerellales; Sordariomycetes) and one alkali-tolerant species from the family Hypocreaceae (Hypocreales; Sordariomycetes). The data obtained showed that DPP4 is the most represented extracellular PSP. Studying the physiology and biochemistry of alkaliphilic and alkali-tolerant fungi is interesting as, on the one hand, it can facilitate the understanding of the mechanisms underlying the adaptation of these organisms to extreme environments and lead to the discovery of new and practically significant enzymes or secondary metabolites. On the other hand, it is crucial to develop measures to prevent biological damage to products by these fungi at alkaline pH. Secretion of PSPs depended on the content of the nutrient medium. Thus, when three species of alkaliphilic fungi of the genus Sodiomyces were grown on three alkaline fermentation media (malt, peptone, casein as nitrogen sources), DPP4 was secreted in all the studied media, but different protein sources stimulated the secretion of the general activity DPP4 to a different degree (Figure 3). When the mycelium of the studied strains was grown on malt extract, it gained mass relatively quickly, but produced a relatively small amount of DPP4. At the same time, when it was grown on a medium with peptone as a nitrogen source, *S. alkalinus* F11, *S. tronii* MAG3, and *S. magadiensis* B39 showed the highest DPP4 activity in terms of dry mycelium mass. The selection of cultivation period for the *S. alkalinus* F11 strain, which was the most effective producer of PSP from the set of selected cultures, showed that the most intensive growth was recorded on 5–18 days, after which the curve of the dry mycelium mass began to reach a plateau. In the lag phase, the studied strain secreted only trace the amounts of PSP, and the most active production of peptidase occurred at the end of the logarithmic growth stage. DPP4 activity increased by day 14 of cultivation from 0 to 7 units/mg of dry mycelium, then gradually decreased to 5 units/mg of dry mycelium by day 24 of growth. DPP4 in all studied alkali-tolerants and in all alkaliphiles, respectively, had a similar pattern of changes in biochemical properties. All the observed differences were related only to the amount of secreted activity. Among the studied alkali-tolerant species, the highest DPP4 activity was found in *Chordomyces antarcticus* M27, which, along with S. alkalinus F11, an effective enzyme producer per dry mycelium among alkaliphiles, was used in further work. Notably, in alkaliphilic representatives of the family (strains of the genus *Sodiomyces*), high DPP4 activity was found not only outside the cells, but also in the cell extract; in alkali-tolerants, no noticeable accumulation of DPP4 in cells was observed.

Comparing the curves of the DPP4 activity dependence on pH in the strains of the alkali-tolerant *C. antarcticus* M27 and the alkaliphilic *S. alkalinus* F11 showed that, in both cases, the DPP4 of the studied species were not able to work in an acidic environment (pH < 5.4) (Figure 4). The highest DPP4 activity in both species was shown in the pH 7–8 region. Although *S. alkalinus* is confined to more alkaline soils, the optimal reaction of the medium for DPP4 of this species was pH 7.3, while the enzyme of *C. antarcticus* with a wider ecological plasticity showed the maximum reaction rate at pH 7.7. At the pH values of more than 8.0, the activity of DPP4 decreased in both species. Our data indicate a slightly alkaline optimum of DPP4 activity in these species, which is consistent with the evidence obtained for homologous proteins of other fungi [24,25]. Notably, in all the experiments, DPP4 of the *S. alkalinus* F11 strain showed a higher enzymatic activity (approximately three times higher) compared with that of *C. antarcticus* M27 (Figure 4).

To survive in conditions of extremely high pH, living organisms have special enzymes that are not highly active at an alkaline pH, but rather alkali-resistant. Measuring the DPP4 activity of F11 and M27 strains after prolonged incubation in a universal buffer of different acidity showed that both enzymes retained hydrolytic activity in a wide pH range. Thus, DPP4 of *S. alkalinus* F11 was stable in the pH range of 5.0–12.0 and retained 23% of its activity after an hour of incubation at pH 13.0. DPP4 of the alkali-tolerant strain M27, in turn, showed a significantly greater relative activity after incubation in an acidic environment (78% vs. 7% after incubation at pH 3.0), but completely denatured at pH 13.0 (Figure 5). Since the studied DPP4 homologs are secreted enzymes, these differences may indicate that *S. alkalinus* F11 adapted to constant alkaline environment, and *C. antarcticum* M27 adapted to a wide range of pH, with an optimum at a neutral pH.

Analysis of the salt resistance of the studied PSP revealed that the activity of extracellular DPP4 strains F11 and M27 in 6 M NaCl solution at optimal pH values was 64% and 63% of the maximum, respectively (Figure 6). Retention of most enzyme activity at extremely high salt concentrations reflects the conditions of natural habitats where these fungi thrive. Thus, determining the stability of DPP4 in excess of NaCl showed that the studied peptidase is a halostable enzyme and retains activity in NaCl solutions close to the saturated one.

To characterize the biochemical properties of the studied DPP4, an enzyme inhibitor analysis was performed using specific peptidase inhibitors. Both *S. alkalinus* F11 and *C. antarcticum* M27 showed a marked inhibition of the enzyme by a specific serine peptidase inhibitor. The addition of 2.5 mM phenylmethylsulfonyl fluoride (PMSF) decreased the DPP4 activity of the F11 strain to 13.7%, and that of M27 strain to 9.3% of the control. The action of DPP IV inhibitors diprotin A and vildagliptin also reduced the studied enzymatic activity. In addition, the strongest inhibitory effect on *S. alkalinus* F11 DPP4 was produced by 0.25 mM diprotin A (reduced activity to 3.2% of the control), and, on *C. antarcticum* M27 DPP4, by 0.25 mM vildagliptin (7.4% of the control). At the same time, competitive PSP inhibitors, such as Ala-Pro (AP)-N-methylformamide (MF), AP-N-methyl-2-pyrrolidone (NMP), and AP-pipecolic acid (Pip), did not decrease DPP4 activity in both cases. The cysteine peptidase inhibitor iodoacetamide and the metalloprotease inhibitor EDTA also had no significant effect on the enzyme (Figure 7). The data obtained indicate that DPP4 belongs to the class of serine peptidases and determine a possible way to control its activity.

## 4. Conclusions

The active development of bioinformatic methods for studying proteins and obtaining hundreds of new sequenced genomes stimulate PSP research in various taxa and facilitate the search for opportunities for the application of these peptidases. To determine the spectrum of PSPs potentially synthesized in fungal cells, a bioinformatic analysis of 42 sequenced genomes was performed, which showed that the genomes of different fungal species differ in the number and set of PSPs; for some enzymes, taxon specificity and adherence to certain ecological and trophic groups were shown. DPP4 was the only PSP present in all the studied genomes, including alkaliphilic and alkali-tolerant species. Such a representation of DPP4 in representatives of various species and genera of fungi may be evidence in favor of the conservativeness and high importance of this peptidase for their vital activity, and indicates the possibility of using fungi as producers of this enzyme. Using a specific substrate and inhibitory analysis, the enzyme was identified in culture filtrates and the properties important for its commercial use were evaluated. The enzymes from the alkaliphilic and alkali-tolerant species had several notable peculiarities, but, in both cases, the DPP4 studied were both active and stable under alkaline conditions and at high salt concentrations, which makes them good candidates for biotechnological applications and bioengineering. Furthermore, DPP4 can be used as a preparation to cleave difficult-to-hydrolyze proline-rich peptides under conditions corresponding to those of the human gut, and may, therefore, be considered as a potential candidate for the enzyme therapy of celiac disease.

## Figures and Tables

**Figure 1 jof-07-00744-f001:**
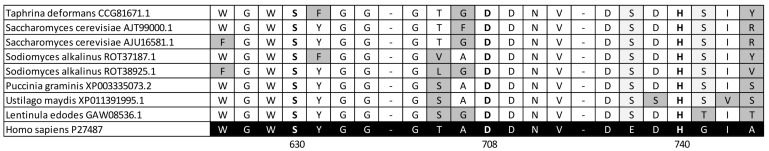
Alignment of conservative DPP4 fragments close to active site in representatives of various subdivisions of higher fungi in relation to human DPP4. Substitutions found in fungi are grayed out, the amino acid residues of DPP4 *Homo sapiens* are highlighted in black, the conservative amino acid triad is highlighted in bold. Amino acid residue numbers are given for DPP4 *H. sapiens*.

**Figure 2 jof-07-00744-f002:**
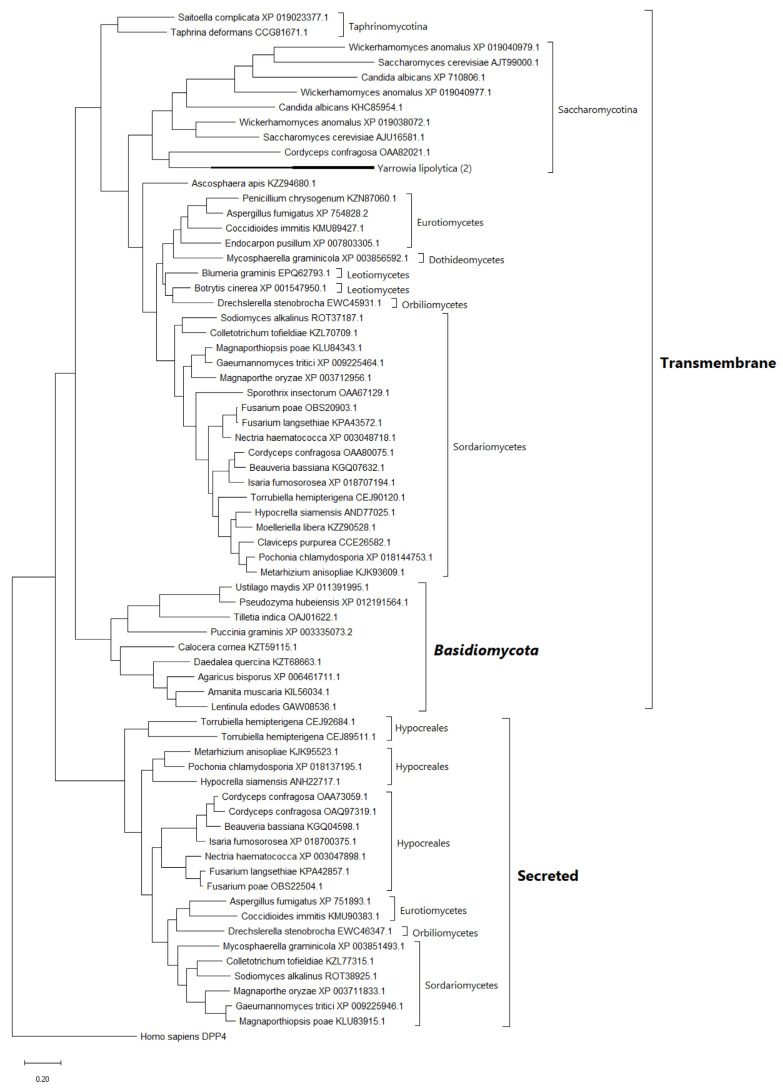
The cladogram of DPP4 homologs constructed by the ML method.

**Figure 3 jof-07-00744-f003:**
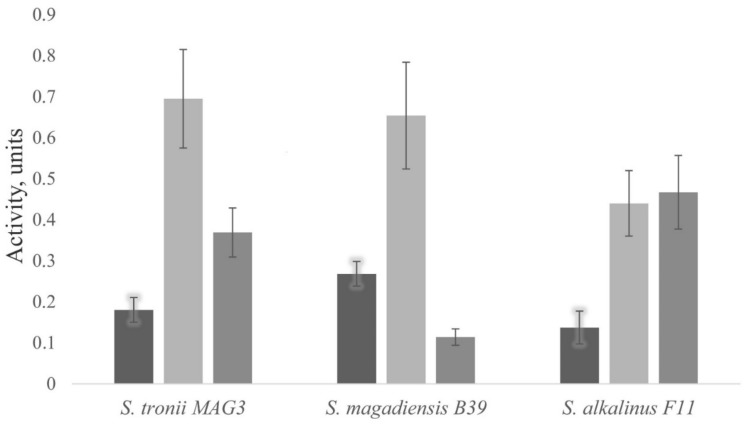
DPP4 activity in the culture fluid of the studied fungi. The medium with wort is black, with peptone is gray, and with casein is dark gray. The limits of the confidence interval at the 95% significance level are given.

**Figure 4 jof-07-00744-f004:**
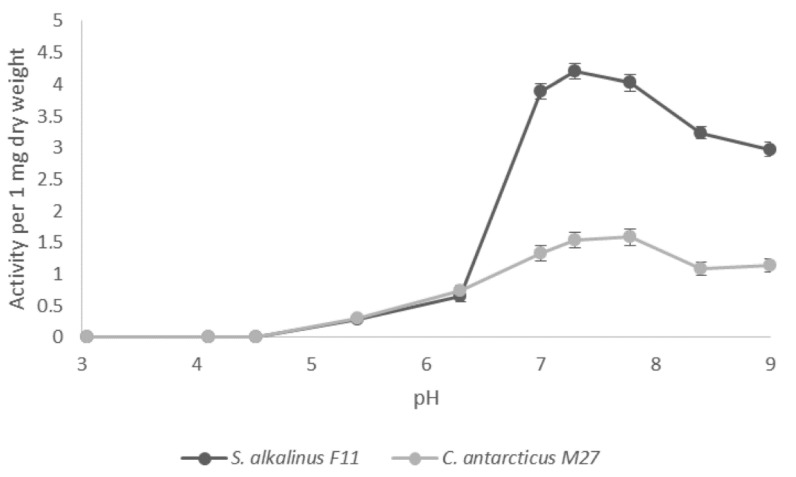
pH dependence of DPP4 activity. The limits of the confidence interval at the 95% significance level are given.

**Figure 5 jof-07-00744-f005:**
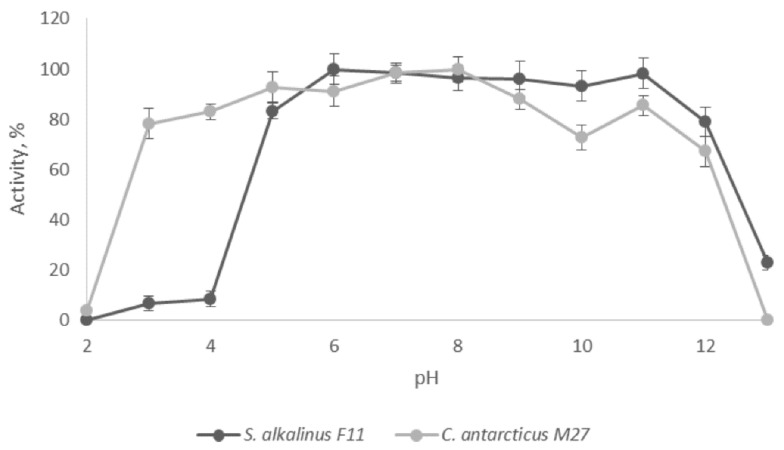
pH stability of DPP4. The limits of the confidence interval at the 95% significance level are given.

**Figure 6 jof-07-00744-f006:**
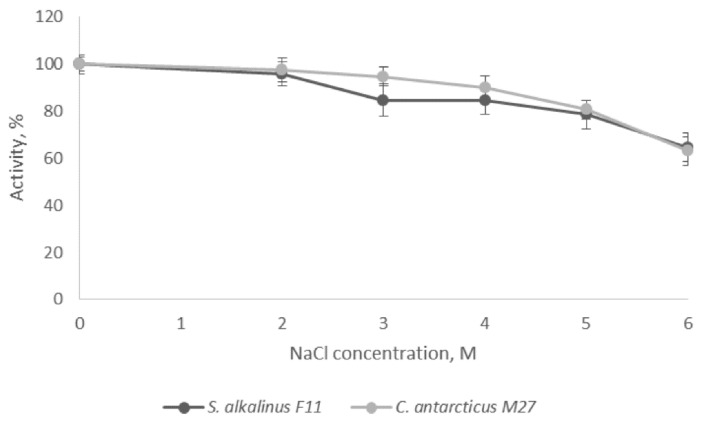
Salt resistance of DPP4. The limits of the confidence interval at the 95% significance level are given.

**Figure 7 jof-07-00744-f007:**
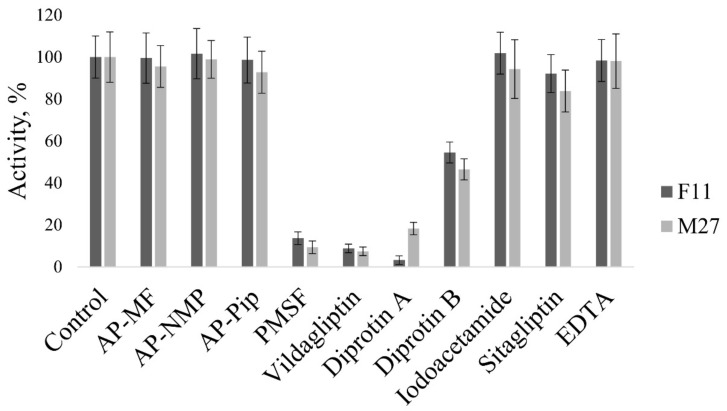
The effect of inhibitors on DPP4. The limits of the confidence interval at the 95% significance level are given.

**Table 1 jof-07-00744-t001:** Profile of PSP homologs in the genomes of higher fungi. The table cells contain the number of homologs.

Taxon	Species	APP1	CND	DPP4	EPR	LAP	PIP1	PIP2	PIP3	POP	XPD
**Taphrinomycotina**	*Taphrina deformans*	1	2	1	1		1				1
	*Saitoella complicata*	1	2	1		1	1		1	1	2
**Saccharomycotina**	*Saccharomyces cerevisiae*	1	2	2							2
	*Wickerhamomyces anomalus*	1	2	3							2
	*Candida albicans*	3	2	2			1				2
	*Yarrowia lipolytica*	1	3	2			1				2
**Pezizomycotina**	*Mycosphaerella graminicola*	1	2	2	10		1	1	1		3
	*Ascosphaera apis*		2	1	1		1				3
	*Aspergillus fumigatus*	1	2	2	3		1				3
	*Penicillium chrysogenium*	1	2	1	1		1		1		3
	*Coccidioides immitis*	1	1	2	2						2
	*Endocarpon pusillum*	1	2	1	1		1				2
	*Blumeria graminis*	1	2	1	2		1		1		1
	*Botrytis cinerea*	1	3	1	2		1	1			3
	*Drechslerella stenobrocha*	1	1	2	4						2
	*Colletotrichum tofieldiae*	1	4	2	3		1	1			3
	*Sodiomyces alkalinus*	1	2	2	1						3
	*Claviceps purpurea*	1	2	1	3		1				3
	*Hypocrella siamensis*			2							
	*Metarhizium anisopliae*	1	2	2	4		1				3
	*Moelleriella libera*	1	2	1	3		1				3
	*Pochonia chlamydosporia*	1	2	2	4		1				4
	*Beauveria bassiana*	2	2	2	4	1	1			1	3
	*Cordyceps confragosa*	3	3	4	8		1				5
	*Isaria fumosorosea*	1	2	2	5		1	1			3
	*Torrubiella hemipterigena*	1	2	3	14		1	1			3
	*Fusarium langsethiae*	1	2	2	2		1				3
	*Fusarium poae*	1	2	2	1		1				3
	*Nectria haematococca*	1	3	2	1		1				4
	*Gaeumannomyces tritici*	1	2	2	5		2				3
	*Magnaporthe oryzae*	1	2	2	6		1				3
	*Magnaporthiopsis poae*	1	3	2	4		1				1
	*Sporothrix insectorum*		2	1	4		2	1			1
**Pucciniomycotina**	*Puccinia graminis*	1	1	1		1			2	1	2
**Ustilaginomycotina**	*Tilletia indica*	1	2	1	1	1			1	1	1
	*Pseudozyma hubeiensis*	2	2	1	1	1			1	1	1
	*Ustilago zeae*	2	2	1	1	1			1	1	1
**Agaricomycotina**	*Amanita muscaria*	2	2	1	3	1			3	1	2
	*Agaricus bisporus*	1	2	1	1	2	1	4	2	1	2
	*Lentinula edodes*	2	2	1	2	1	1	4	1	1	2
	*Daedalea quercina*	2	2	1	3	1		8	1	1	2
	*Calocera cornea*	2	2	1	3	1		7	1	1	2

APP1—X-Pro aminopeptidase; CND—cytosolic nonspecific dipeptidase; DPP4—dipeptidyl peptidase; EPR—acid prolyl endopeptidase from *As. niger*; LAP—leucine aminopeptidase; PIP1—prolyliminopeptidase from *A. sobria*; PIP2—prolyliminopeptidase from *L. delbrueckii*; PIP3—prolyliminopeptidase from *D. carota*; POP—prolyl oligopeptidase; XPD—prolidase.

**Table 2 jof-07-00744-t002:** Cultures used for analysis of activity.

Species	Strains	The Gathering Place	Relation to pH
*Acrostalagmus luteoalbus*	V205 (CBS 137625)	Kulundinskaya steppe; lake	alkalitolerant
*Chordomyces antarcticus*	M27 (CBS 120045)	Kulundinskaya steppe; lake	alkalitolerant
*Sodiomyces alkalinus*	F11 (CBS 110278)	Mongolia; lake	alkalophile
*Sodiomyces magadiensis*	MAG5 (CBS 142933)	Kenya, soda soil on the shore of the lake Magadi	alkalophile
	B39 (CBS 142937)	Russia; saline soil on the shore of the lake Baskunchak	alkalophile
*Sodiomyces tronii*	MAG3 (CBS 137620)	Kenya, lake Magadi; the shore of the lake	alkalophile
*Verticillium zaregamsianum*	V201 (CBS 137621)	Transbaikalia; lake	alkalitolerant

## Data Availability

The data were presented and discussed at Congress of Biochemists of Russia (2019), International Multiconference “Bioinformatics of Genome Regulation and Structure/Systems Biology (2019, 2020), International Scientific Conference Lomonosov” (2019, 2020) and are reflected in the conference materials.

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
