# Peer review of "Proline-Specific Fungal Peptidases: Genomic Analysis and Identification of Secreted DPP4 in Alkaliphilic and Alkalitolerant Fungi"

_jof, 2021, doi:10.3390/jof7090744_

Round 1
Reviewer 1 Report
Review for:
Proline-specific fungal peptidases: genomic analysis and identification of secreted DPP4 in alkaliphilic and alkalitolerant fungi
Alkin and coworkers performed a bioinformatic analysis of fungal proline-specific peptidases with a focus on dipeptidyl peptidase 4 (DPP4) from alkaliphilic or alkaline tolerant fungi. As outlined in the introduction proline-specific peptidase might have the potential to help patients with celiac disease. The identification of enzymes which remain functional in alkaline conditions might have further technological benefits. The presence of proline-specific peptidases in alkaline-tolerant and alkaliphilic fungi was clearly established by the authors. It was interesting to read that DPP4 can occur as a transmembrane and a secreted version as two distinct paralogs at least for some fungi (for example ascomycetes). I was somewhat disappointed by the biochemical analysis of DPP4 for a few selected alkaliphilic or alkaline tolerant fungi. At first 5 different fungal species are listed as being investigated, but data is then shown for only three (Figure 3) and later only two species (Figure 4-7). It remains unclear how these 5,3, or 2 species were selected. I recommend to highlight the selected species in the tree shown in Figure 2 and to make a stronger connection between part 1 (bioinformatics) and part 2 (biochemical analysis). Since the existence of paralogs for DPP4 was mentioned, I had expected that the biochemical analysis would include a two-pronged sample preparation with a separation of the secreted proteins (studying the growth medium) and a preparation of the remaining washed fungi with a cell lysis procedure using a detergent to obtain the transmembrane paralog of DPP4. If a comparison of the paralogs is outside the scope of this manuscript, please provide a strong motivation for the narrower focus on secreted forms of DPP4. The study presented by Alkin and coworkers should be of interest for many readers working on either fungi or enzymes (especially in biotechnological applications). So, I recommend to publish the study after revising the overall presentation. Below is a list of my concerns:
Major concerns:
- Please improve the presentation of the biochemical analysis with a stronger more transparent motivation on why which fungal species was selected for the presented experiments.
- Please provide a better connection between the findings of the bioinformatic analysis and the design of the biochemical experiments.
- The manuscript would benefit from a conclusion to bring the two parts (bioinformatic and biochemical experiments) into one coherent picture.
Minor concerns:
Lines 100-105: The abbreviations of the peptidases are somewhat hidden in this long list of database entries. It would be easier for the reader to present the specific peptidases and their abbreviations earlier.
Table1: The abbreviations of the enzyme names (see point above) could also be incorporated into a footnote of this table.
Lines 138-149: Please turn these lists (looks like a lab manual or a protocol) into regular sentences/paragraph format.
Line 161: Please provide information on the chemical composition of the Universal Buffer (UB).
Line 270: Is there a significance to calculating the isoelectric points that can be used to strengthen an argument for alkaline tolerance? If yes, please motivate the presentation and calculation of the isoelectric point.
Line 291: an extra underline symbol between two sentences. (a typo)
Figure 2: Why is Basidiomycota in bold?
Figure 3: The numbers on the y-axis should not have a comma (use a period).
References 18-22: If possible look for the “How to Cite” instructions for programs like BLAST and provide the last day of access.
Author Response
We are grateful to the reviewer for the careful reading of the article and making comments. Below we note what has been done:
Please improve the presentation of the biochemical analysis with a stronger more transparent motivation on why which fungal species was selected for the presented experiments.
The same set of PSPs was characteristic of all the biochemically studied alkaliphiles and alkalitolerants. In all the experiments, the alkaliphilic strains demonstrated greater enzymatic activity in comparison with the alkalitolerant fungi, which may be explained by different ecological strategies of the studied species: alkaliphilic fungi have to survive in a narrower pH range than the alkalitolerant ones. DPP4 in all studied alkalitolerants and in all alkaliphiles, respectively, had a similar pattern of changes in biochemical properties. All the observed differences were related only to the amount of secreted activity. One of the effective producer of proteases per dry mycelium among alkaliphilic fungi was S. alkalinus F11, among alkalitolerant fungi – C. antarcticum M27. In order not to complicate the drawings, we used them as representatives to compare the characteristic properties of alkaliphiles and alkalitolerants. Explanation inserted in the text.
Please provide a better connection between the findings of the bioinformatic analysis and the design of the biochemical experiments.
The aim of this work was to evaluate the potential of fungi in the production of various types of PSP using a bioinformatic analysis of complete genomes, followed by the biochemical identification of PSPs in the cultures of fungi, as well as to characterize properties contributing to their further use. The genomes of different fungal species differed in the number and set of PSP. DPP4 was the only PSP represented in all the studied genomes. Such a representation of DPP4 in representatives of various species and genera of fungi may be evidence in favor of the conservativeness and high importance of this peptidase for their vital activity and indicate the possibility of using fungi as producers of this enzyme. Using a specific substrate and inhibitory analysis, the enzyme was identified in culture filtrates and the properties important for its commercial use were evaluated.
The manuscript would benefit from a conclusion to bring the two parts (bioinformatic and biochemical experiments) into one coherent picture.
Conclusion added to the text of the article.
Lines 100-105: The abbreviations of the peptidases are somewhat hidden in this long list of database entries. It would be easier for the reader to present the specific peptidases and their abbreviations earlier.
An insertion was made in section 2.2 "Materials and methods". Analyzed peptidases:
prolyl oligopeptidase (POP), prolidase (XPD), fibroblast activation protein (FAP), dipeptidyl peptidases (DPP)4, DPP6, DPP8, DPP9, DPP10, X-Pro aminopeptidases (APP)1 and APP2, acid prolylendopeptidase (EPR) from Aspergillus niger, prolyliminopeptidases from Aeromonas sobria (PIP1), from Lactobacillus delbrueckii (PIP2) and from carrot (Daucus carota, PIP3), leucine aminopeptidase (LAP) and cytosolic non-specific dipeptidase (CND).
Table1: The abbreviations of the enzyme names (see point above) could also be incorporated into a footnote of this table.
Footnote to Table 1: APP1 - X-Pro aminopeptidase; CND - cytosolic non-specific dipeptidase; DPP4 - dipeptidyl peptidase; EPR - acid prolylendopeptidase from As. niger; LAP - leucine aminopeptidase; PIP1 - prolyliminopeptidase from A. sobria; PIP2 – prolyliminopeptidase from L. delbrueckii; PIP3 - prolyliminopeptidase from D. carota; POP - prolyl oligopeptidase; XPD – prolidase.
Lines 138-149: Please turn these lists (looks like a lab manual or a protocol) into regular sentences/paragraph format.
Reworked in accordance with the reviewer comment. "The composition of the nutritional component, in addition to the yeast extract, included either malt extract (15° on the Balling scale), or casein, or peptone."
Line 161: Please provide information on the chemical composition of the Universal Buffer (UB).
Universal Buffer (UB) consists of a mixture of 0.04 M boric, 0.04 M phosphoric and 0.04 M acetic acids, which was titrated to the desired pH using 0.2 M NaOH.
Line 270: Is there a significance to calculating the isoelectric points that can be used to strengthen an argument for alkaline tolerance? If yes, please motivate the presentation and calculation of the isoelectric point.
We considered isoelectric points only as individual characteristics of the found enzymes, along with the number of domains, signal sequences, and functional groups of the active center. However, one can refer to articles where the relationship between isoelectric points and the pH optimum or pH stability under certain conditions is not found (Alexov 2004. Eur. J. Biochem. 271, 173–185; Talley and Alexov 2010. Proteins 78, 2699–2706).
Line 291: an extra underline symbol between two sentences. (a typo)
Eliminated.
Figure 2: Why is Basidiomycota in bold?
Bold removed
Figure 3: The numbers on the y-axis should not have a comma (use a period).
Commas are replaced with periods
References 18-22: If possible look for the “How to Cite” instructions for programs like BLAST and provide the last day of access.
Added last access date to program links
Reviewer 2 Report
This is an interesting paper on determination of fungal species producing proline-specific peptidases and itheir relation to popular drug target human DPP IV. First they used a bioinformatic tools and then studied DPP IV-like activity in culture media of five fungal species, followed by characterization of two activities by standard biochemical techniques (pH, salt concentration dependence of activity and sensitivity on action of certain inhibitors). Paper is well written and presented. I would like to check it carefully bacuse I have found some editorial errors but certainly not all of them. These are:
1./ should be "radiation [1-4]" (line 32);
2./ some names of producers in paragraph 2.1. are given in wrong font;
3./ iodoacetamidand USA are in Russian in the same paragraph;
4./ I would like to propoe to add names of all the enzymes as a footnote under the Table 1. It is difficult to follow the Table with names given in Experimental or text body;
5./ remove "Fig.3" and "Fig 4" from Figures 3 and 4. I would like also to propose to combine these figures in one using A and B notation;
Finally, part of the disscussion devoted to characterization of DPP IV-like activity of enzymes from C. antarcticum and S. alkalinusis really badly written and needs ordering. For example, a fragment devoted to action of specific inhibitors of DPP IV could be like following " action of DPP IV inhibitors diprotin A and vildagliptin decreased the studied enzymetic activity. And thus,...(and here some specific data)"
Author Response
Response to Reviewer 2 Comments
We are grateful to the reviewer for the careful reading of the article and making comments. Below we note what has been done:
1./ should be "radiation [1-4]" (line 32);
Fixed.
2./ some names of producers in paragraph 2.1. are given in wrong font;
Fixed
3./ iodoacetamidand USA are in Russian in the same paragraph;
Fixed
4./ I would like to propoe to add names of all the enzymes as a footnote under the Table 1. It is difficult to follow the Table with names given in Experimental or text body;
Footnote to Table 1: APP1 - X-Pro aminopeptidase; CND - cytosolic non-specific dipeptidase; DPP4 - dipeptidyl peptidase; EPR - acid prolylendopeptidase from As. niger; LAP - leucine aminopeptidase; PIP1 - prolyliminopeptidase from A. sobria; PIP2 – prolyliminopeptidase from L. delbrueckii; PIP3 - prolyliminopeptidase from D. carota; POP - prolyl oligopeptidase; XPD – prolidase.
5./ remove "Fig.3" and "Fig 4" from Figures 3 and 4. I would like also to propose to combine these figures in one using A and B notation;
Extra numbers removed from all drawings. We have left the drawings separately to simplify their placement in the text
Finally, part of the disscussion devoted to characterization of DPP IV-like activity of enzymes from C. antarcticum and S. alkalinusis really badly written and needs ordering. For example, a fragment devoted to action of specific inhibitors of DPP IV could be like following " action of DPP IV inhibitors diprotin A and vildagliptin decreased the studied enzymetic activity. And thus,...(and here some specific data)"
Corrections have been made to the text. Added a conclusion.
Round 2
Reviewer 1 Report
The authors addressed my concerns from the first review. I have one last recommendation: To use the subheading 5. Conclusion for the last paragraph of the manuscript, so that it is easier for reader to identify.
Author Response
Dear reviewer, we have somewhat expanded the conclusion and used the subtitle 4 for it.